materials science

biomaterial, composite, glass, ion-release, poly(lactic-*co*-glycolic acid), calcium carbonate

**Authors for correspondence:**
Naoki Osada
e-mail: osada@orthorebirth.com
Toshihiro Kasuga
e-mail: kasuga.toshihiro@nitech.ac.jp

# Tuning of ion-release capability from bio-ceramic-polymer composites for enhancing cellular activity

Naoki Osada[1,2], Arisa Terada[1], Hirotaka Maeda[1], Akiko Obata[1], Yasutoshi Nishikawa[2] and Toshihiro Kasuga[1]

[1]Nagoya Institute of Technology, Gokiso cho, Showa ku, Nagoya 466-8555, Japan
[2]ORTHOREBIRTH Co. Ltd, 15-3-303 Chigasaki-Chuo, Tsuzuki-ku, Yokohama 224-0032, Japan

NO, 0000-0003-2424-4678; HM, 0000-0003-2841-6211

In our previous study, we investigated the synergetic effects of inorganic ions, such as silicate, $Mg^{2+}$ and $Ca^{2+}$ ions on the osteoblast-like cell behaviour. $Mg^{2+}$ ions play an important role in cell adhesion. In the present study, we designed a new composite that releases a high concentration of $Mg^{2+}$ ions during the early stage of the bone-forming process, and silicate and $Ca^{2+}$ ions continuously throughout this process. Here, $40SiO_2$–$40MgO$–$20Na_2O$ glass (G) with high solubility and vaterite-based calcium carbonate (V) were selected as the source of silicate and $Mg^{2+}$ and $Ca^{2+}$ ions, respectively. These particles were mixed with poly(lactic-*co*-glycolic acid) (PLGA) using a kneading method at 110°C to prepare the composite (G-V/PLGA, G/V/PLGA = 4/56/40 (in weight ratio)). Most of the $Mg^{2+}$ ions were released within 3 days of immersion at an important stage for cell adhesion, and silicate and $Ca^{2+}$ ions were released continuously at rates of 70–80 and 180 ppm d$^{-1}$, respectively, throughout the experiment (until day 7). Mouse-derived osteoblast-like MC3T3-E1 proliferated more vigorously on G-V/PLGA in comparison with V-containing PLGA without G particles; it is possible to control the ion-release behaviour by incorporating a small amount of glass particles.

# 1. Introduction

Tissue engineering combining three elements: namely, cells, scaffolds and growth factors, has been attracting attention in recent years as a method for repairing tissues damaged from injury or disease [1]. Examples of the growth factor include

proteins typified by bone morphogenetic protein, amino acids and inorganic ions. These proteins and amino acids have many problems such as the need to be supported and released without a loss in their activity [2]. Inorganic ions are attracting attention as an inducing initiator because there is no denaturation during material synthesis. Silicate ions are known as essential elements for metabolic processes involved in bone formation [3]. Silicate ions have been reported to promote the proliferation, differentiation and calcification of osteoblast-like cells [4–10]. $Ca^{2+}$ ions are significant components in bone and are an indispensable element for bone formation. It has been reported that $Ca^{2+}$ ions also promote the proliferation, differentiation and calcification of osteoblast-like cells [11,12].

Poly(L-lactic acid) (PLLA) has been reported as a biodegradable polymer for bone repair [13–15]. Our group focused on silicate and calcium ions, and prepared siloxane containing vaterite (SiV) for their release [16]. PLLA composites containing SiV have been reported to promote bone formation [17], and have been successfully shaped into a cotton-wool-like form through electrospinning [18], namely, the adhesion of osteoblast-like cells starts around their cross-linking sites, and the cells penetrate between the fibres [19]. Our group has also developed a new type of cotton-wool-like, fibrous material containing SiV and β-tricalcium phosphate (β-TCP) particles. This material has shown excellent bone formation in *in vivo* tests using New Zealand white rabbits [20]. Because the PLLA is present as a matrix phase in the fibre, SiV on the surface dissolves during an early stage, releasing calcium and silicate ions. β-TCP particles were expected to be dissolved extremely slowly in the body, providing positive effects within a relatively long period after implantation. Their dissolution was controlled slowly through the decomposition of the PLLA matrix. PLLA, which does not exhibit rapid degradability, can maintain the fibrous shape during bone regeneration.

The degradability of the polymer matrix phase can be controlled using poly(lactic-co-glycolic acid) (PLGA) by changing the ratio of lactic acid and glycolic acid [21]. PLGAs, which are basically hydrophobic, show a swelling ability when placed in water, originating from their high degradability [22,23]. Our group reported that the PLGA composites containing calcium carbonate particles showed a continuous release of calcium ions [24]. When PLGA is compounded with an inorganic material, that is, the rapid hydrolysis of PLGA is expected to induce the effective release of therapeutic ions such as $Ca^{2+}$ and silicate ions. PLGA might be an excellent matrix polymer for preparing a composite with a higher osteogenesis promoting effect.

$Mg^{2+}$ ions are also essential elements for bone metabolism [25]. It has been reported that, among osteoblast-like cell mineralization processes, such ions play an important role in adhesion, acting on osteoblast integrins [25]. In recent years, the influence of the combination of multiple ions on the cell behaviour has been investigated [26–28]. In our group, when osteoblast-like cells were cultured in a medium in which the particular concentrations of $Ca^{2+}$, silicate and $Mg^{2+}$ ions were increased, their proliferation, differentiation and mineralization were promoted [29–31]. A typical concentration for enhancing the mineralization is Ca of 140 ppm, Si of 58 ppm, Mg of 71 ppm and P of 36 ppm (Ca of 78 ppm, Si of 0 ppm, Mg of 28 ppm and P of 40 ppm in a normal alpha minimum essential medium (α-MEM)).

The authors considered that, to induce high osteoblast activities, it is necessary to continuously supply $Mg^{2+}$ ions, silicate ions during the initial stage and a large amount of $Ca^{2+}$ ions after implantation of the materials. Composites using PLGA will be a key to providing a therapeutic ion-release capability.

Our group prepared a composite material using PLGA and SiV, and investigated its ion-releasing behaviour [32]. Although the continuous release of calcium ions from SiV/PLGA over a long period was observed, most of the silicate ions were released within 1 day. The water-uptake ability of PLGA is considered to form an ion-releasing pathway [24]. Amorphous siloxane bonds weakly around the primary particles of vaterite, which easily dissociates to be released in an aqueous solution [32,33].

Bioglass® (45S5, 46.1$SiO_2$–26.9$CaO$–24.4$Na_2O$–2.6$P_2O_5$ in mol% [34–36], developed by Hench) is an alkaline that immediately occurs within the vicinity of the surface after implantation in the body because calcium and sodium ions are exchanged with protons in water, forming a silanol group when in contact with bodily fluids. As a result, soluble silica of $Si(OH)_4$ is produced, a portion of which condenses again to form a silica gel phase. Some of the soluble silica is continuously released. Silicate glasses containing MgO are promising sources for the release of silicate and magnesium ions.

In this study, to control the release of inorganic ions in enhancing bone formation, $SiO_2$–$MgO$–$Na_2O$ glass and calcium carbonate (vaterite) particles were embedded in a PLGA matrix phase. The objectives were to examine the ion-releasing behaviour and cell proliferation of the prepared composite materials and to discuss their possibility as a new bone regeneration material.

(a)

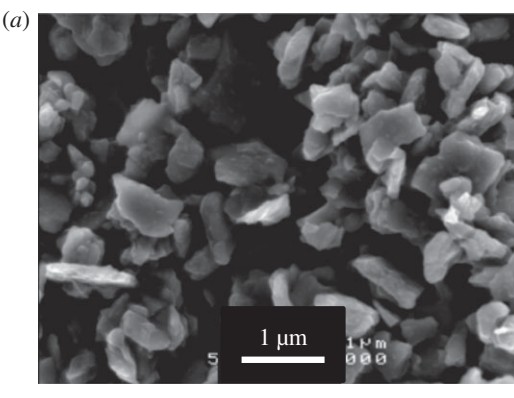

(b)

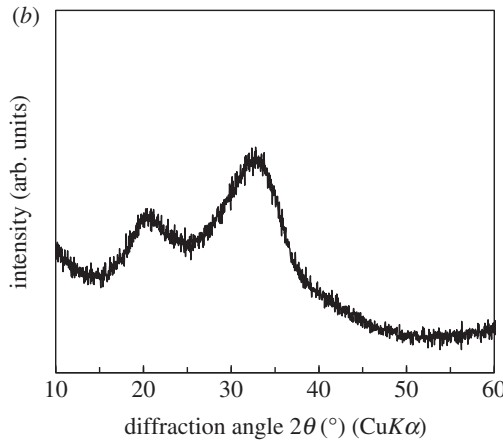

**Figure 1.** (a) SEM image and (b) XRD pattern of G particles.

# 2. Material and methods

## 2.1. Preparation of $SiO_2$–$MgO$–$Na_2O$ glass

A batch mixture of nominal compositions of $40SiO_2$–$40MgO$–$20Na_2O$ in mol% was prepared using reagent-grade chemicals such as $SiO_2$, $MgO$ and $Na_2CO_3$ (Kishida Chemical, Japan). The mixture was melted at 1500°C for 30 min in a platinum crucible in air and then quenched on stainless steel using an iron-pressing method, resulting in the formation of plate-likes glass pieces. The resulting glasses are denoted as 'G'. The glasses were crushed roughly using an alumina mortar and then pulverized using ball-milling for 24 h with 400 g of zirconia balls of 3 mm in diameter and 30 g of glass with 100 ml of methanol.

Figure 1a shows an image of the G particles using scanning electron microscopy (SEM) (JSM-6301F, JEOL, Japan). A conductive coating treatment was carried out prior to the observation using an osmium coater (NEOC Neo Osmium Coater, Meiwafosis Co., Ltd, Japan). The glass particle sizes were approximately 1 μm or less. As shown in figure 1b, halo peaks were observed in an X-ray diffraction (XRD; X'pert, Philips, the Netherlands: $CuK\alpha$, 45 kV, 40 mA) pattern of the particles; G was concluded to be glassy particles. In this pattern, two halo peaks were shown; this glass may contain two different networks.

The structure of the prepared glass sample was measured using laser Raman spectroscopy with an Nd: YAG laser (NRS-3300, 532 nm, JASCO Co., Japan). The structure of silicon in the glass was examined through magic-angle-spinning nuclear magnetic resonance spectroscopy (MAS-NMR) (JNM-ECA600II, JEOL, Japan) at a resonance frequency of 119.24 MHz, operating at pulses of 5.0 μs and at a recycle delay of 120 s using an 8.0 mm zirconia rotor. The sample spinning speed was 6 kHz. In addition, 4-dimethyl 4-silapentane sulfonate sodium (1.534 ppm) was used as a reference material of $^{29}Si$.

## 2.2. Preparation of vaterite particles

Calcium carbonates consisting predominantly of vaterite were prepared using a carbonation process in methanol [37]. $CO_2$ gas was blown for 3 h at a flow rate of 300 ml min$^{-1}$ into a suspension consisting of 7.0 g of $Ca(OH)_2$ in 180 ml of methanol at 0°C in a Pyrex beaker. The resulting slurry was dried at 70°C in air, resulting in the formation of fine-sized powders. Figure 2 shows an SEM photograph of the powders. The surface area of the calcium carbonates obtained was determined to be 40 m$^2$ g$^{-1}$ through a nitrogen gas sorption analysis. Secondary particles of 0.5–1 mm in diameter were formed, and an agglomeration of primary particles of 20–100 nm in diameter was observed. XRD analysis ($CuK\alpha$, 45 kV, 40 mA) showed that the calcium carbonate consists predominantly of vaterite with a trace amount of calcite (denoted as V hereafter).

## 2.3. Preparation of PLGA composites

To prepare a composite containing G and V, PLGA (75% lactide/25% glycolide) (PURASORB®, Corbion Purac, The Netherlands) with an average molecular weight ($M_w$) of 102 kDa was used. PLGA was melted

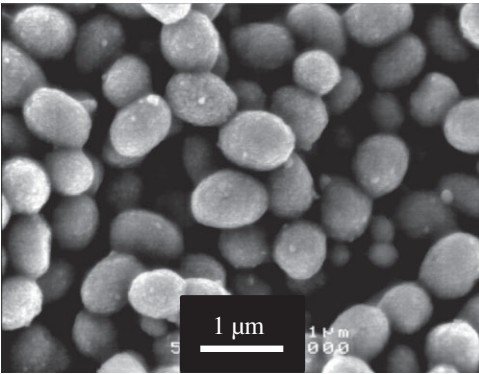

**Figure 2.** SEM image of V particles.

**Table 1.** Compositions of the prepared composites.

| sample code | composition (wt%) |
|---|---|
| G-V/PLGA | G: 4, V: 56, PLGA: 40 |
| V/PLGA | V: 60, PLGA: 40 |
| TCP/PLGA | β-TCP: 60, PLGA: 40 |

at 135°C in a kneader, and then mechanically blended with G and V particles for 10 min. The resulting composite is denoted as G-V/PLGA. For comparison, composites consisting of V and PLGA (denoted by V/PLGA) and consisting of β-TCP and PLGA (denoted by TCP/PLGA) were also prepared using the same procedure as G-V/PLGA. The composition of each sample is shown in table 1.

Each composite was dissolved in chloroform. The composite slurry was cast in a Teflon® vessel and dried at room temperature to form a film. Film-like samples were then used for the following evaluation tests.

## 2.4. Measurement of ion release from composites

The amounts of ions released from the G-V/PLGA films (14 mm in diameter and $160 \pm 20\,\mu m$ in thickness) were evaluated by immersing them in 1 ml of a culture medium ($\alpha$-MEM) containing 10% fetal bovine serum (FBS) and incubating at 37°C in a humidified atmosphere of 95% air with 5% $CO_2$ for 7 days. The medium was changed after 1 day of culturing and then changed every other day.

An XRD analysis (Cu$K\alpha$, 45 kV, 40 mA) of G-V/PLGA was conducted to examine the change in crystal phase before and after immersion in $\alpha$-MEM.

The supernatant solution after immersion was diluted 10 times with distilled water, and the concentration of each ion in the solution was measured using inductively coupled plasma atomic emission spectrometry (ICP-AES) (ICPS-7000, Shimadzu, Japan).

To examine the charge of the ion distribution in G-V/PLGA, the composite was immersed in $\alpha$-MEM for 7 days, and to prepare the sample for observation with a scanning transmission electron microscope (STEM) (JEM-2100F, JEOL, Japan) equipped with an energy dispersive X-ray spectrometer, the composite was processed using a focused ion beam processing observation apparatus (FIB) (EM-9320FIB, JEOL, Japan). An element mapping image of a section of the sample was obtained.

## 2.5. Cell culture test

PLGA, G-V/PLGA, V/PLGA and TCP/PLGA films of 14 mm in diameter and $160 \pm 20\,\mu m$ in thickness were prepared for the cell culture tests. They were sterilized using ethylene oxide gas. Mouse osteoblast-like cells (MC3T3-E1 cells) were seeded onto the films in 24-well plates at a density of 30 000 cells well$^{-1}$. The $\alpha$-MEM containing 10% FBS was used as the culture medium, and the cells were cultured at 37°C in 5% $CO_2$ for 7 days. The medium was changed after 1 day of culturing and then changed every other day.

The number of MC3T3-E1 cells was evaluated after treatment using a Cell Counting Kit-8 (Dojindo, Japan). One reagent used in the kit is a water-soluble tetrazolium salt, which is reduced by

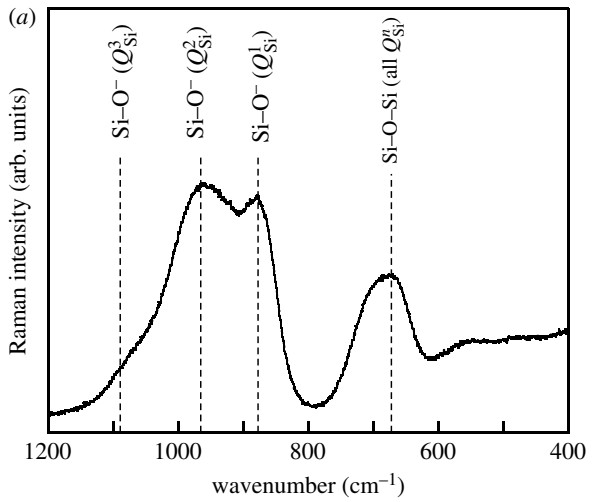
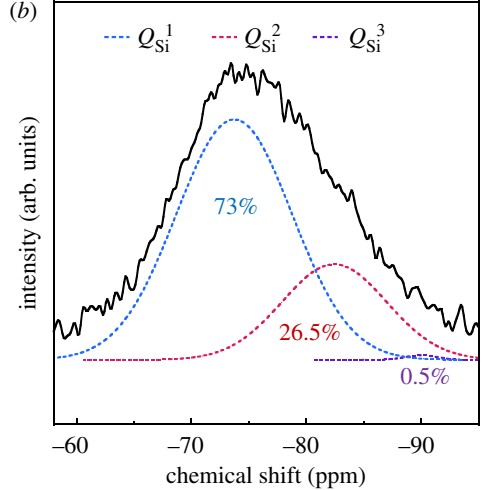

**Figure 3.** (a) Laser Raman and (b) $^{29}$Si MAS-NMR spectra of G particles.

dehydrogenases in live cells to generate a water-soluble formazan. The water-soluble formazan reaches the maximum absorption at approximately 460 nm. The numbers of live cells in the samples were counted by measuring the absorbance of the resulting medium at 450 nm using an absorption microplate reader (Sunrise Remote, TECAN Japan, Japan). The concentration of ions in the medium after culturing was measured using ICP-AES.

# 3. Results

## 3.1. Structure of G particles

Figure 3a shows the laser Raman spectrum of the G particles. The peaks derived from $Q_{Si}^1$, $Q_{Si}^2$ and $Q_{Si}^3$ were reported to appear at approximately 650 and 880 cm$^{-1}$, 650 and 970 cm$^{-1}$ and 1080 cm$^{-1}$, respectively [38–40]. The $Q_{Si}^3$ peak was unclear.

Figure 3b shows the spectrum of 29Si MAS-NMR and the deconvoluted peaks using a Gaussian function. The peaks derived from $Q_{Si}^1$, $Q_{Si}^2$ and $Q_{Si}^3$, which were confirmed to appear at −73, −82 and −91 ppm, respectively, were observed [39].

The peak-integrated ratio of $Q_{Si}^1/Q_{Si}^2$ was estimated to be approximately 73/27. As a result, the $Q_{Si}^1$ unit was concluded to be a dominant structure without the $Q_{Si}^3$ unit.

## 3.2. Ion release from G-V/PLGA

Figure 4 shows the XRD patterns of G-V/PLGA before and after immersion in α-MEM. The peaks derived from vaterite were confirmed. A weak peak corresponding to calcite appearing in the pattern was confirmed.

Figure 5 shows the ion release behaviour from G-V/PLGA in α-MEM. The Mg$^{2+}$ and Ca$^{2+}$ ions in the media were 28 and 160 ppm, respectively. Silicate ions were released continuously (approx. 70–80 ppm) from the composite during the 7-day period. Approximately 74% of the Mg$^{2+}$ ions were released within 1 day, and approximately 90% were released within 3 days. Ca$^{2+}$ ions were released continuously, similarly as the silicate ions.

Figure 6 shows element mapping images of the cross-section of G-V/PLGA around its surface before and after the 7 days of immersion. In the images prior to immersion, V-derived Ca and G-derived Si were observed. Note that the amount of G was only 4%. The G particles were sparsely embedded in the PLGA matrix phase. In the images taken after 7 days of immersion, V-derived Ca was clearly observed; almost no change in the distribution state was observed before or after immersion. By contrast, from the images taken after 7 days of immersion, no G particles were observed inside the material, and Si and P appeared within the vicinity of the sample surface on the V particles.

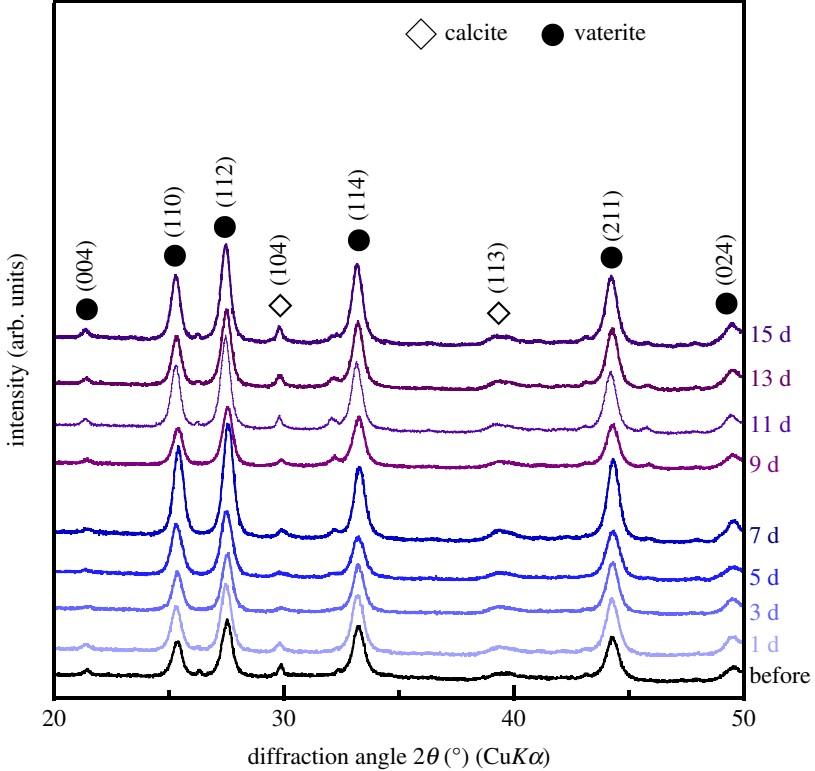

**Figure 4.** XRD patterns of G-V/PLGA before and after immersion in α-MEM.

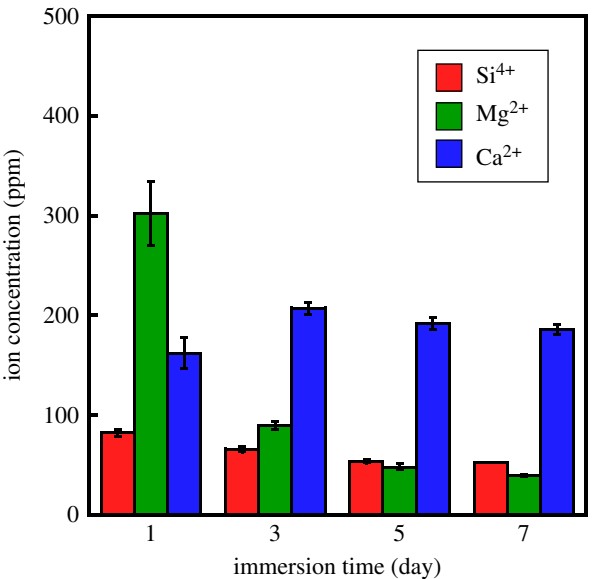

**Figure 5.** Ion-release amounts from G-V/PLGA in α-MEM. An immersion time of '1' means '0–1' day, whereas '3,' '5,' and '7' indicate '2–3,' '4–5,' and '6–7' days, respectively.

## 3.3. Proliferation of MC3T3-E1 cells

Figure 7 shows the numbers of MC3T3-E1 cells after culturing on PLGA, TCP/PLGA, V/PLGA and G-V/PLGA for 7 days. In all samples, an increase in the number of cells was observed. G-V/PLGA showed a significantly higher number of viable cells compared with PLGA.

Figure 8 shows the amounts of ions released in α-MEM from TCP/PLGA, V/PLGA and G-V/PLGA after the culturing. In the case of TCP/PLGA, almost no changes in the amounts of $Mg^{2+}$ and $Ca^{2+}$ ions in the media after the culturing could be observed. In the case of V/PLGA, the release of a large of amount

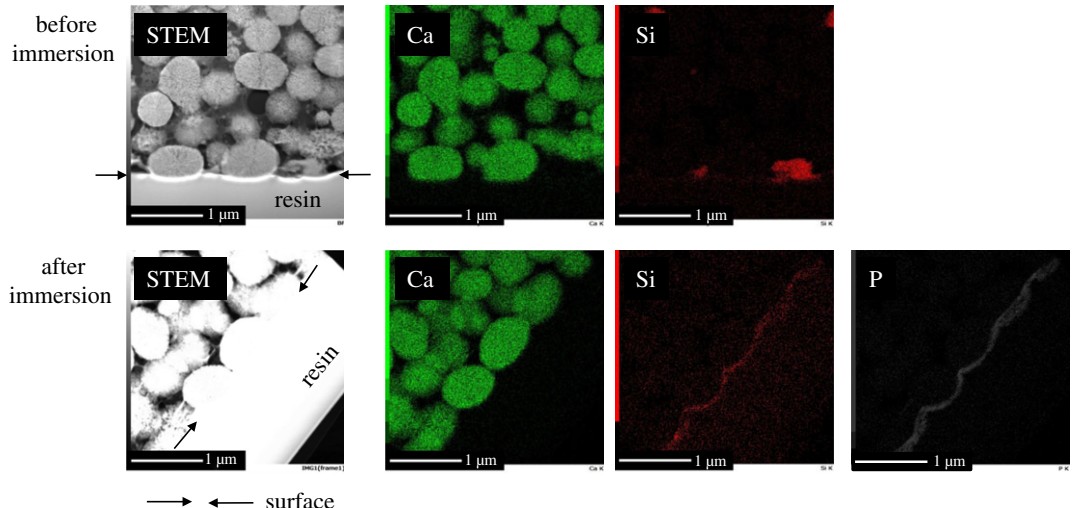

**Figure 6.** Element mapping image of cross-sectional G-V/PLGA before and after 7 days of immersion in $\alpha$-MEM. The samples were prepared by FIB processing after being embedded in 'resin'. The arrow indicates the surface of the composite.

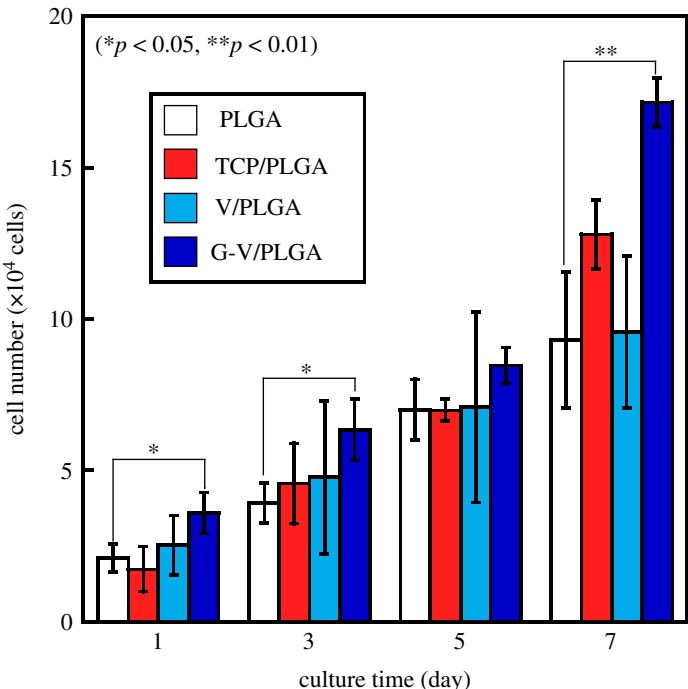

**Figure 7.** Numbers of MC3T3-E1 cells after culturing on PLGA, TCP/PLGA, V/PLGA and G-V/PLGA for 7 days.

of $Ca^{2+}$ ions was observed within 1 day, and the amounts released decreased with an increase in the culture time. In the case of G-V/PLGA, most of the $Mg^{2+}$ ions were released within 3 days, and silicate and $Ca^{2+}$ ions were continuously released as well, as shown in figure 5.

## 4. Discussion

G included approximately 70% of $Q_{Si}^1$ and 30% of $Q_{Si}^2$. In general, it may be difficult to prepare glasses containing almost no $Q_{Si}^2$, $Q_{Si}^3$ or $Q_{Si}^4$. Figure 1b may imply two types of network structures in the glass. It has been reported that, in $49.5SiO_2$–$1.1P_2O_5$–$23.0MgO$–$26.4Na_2O$ glass, approximately 85% of MgO act as network modifiers and approximately 15% as network formers, $MgO_4$ units [41]. Because MgO, which is an intermediate oxide, is contained in large amounts, some of them are considered to act as a member in glass network formers facilitating the vitrification.

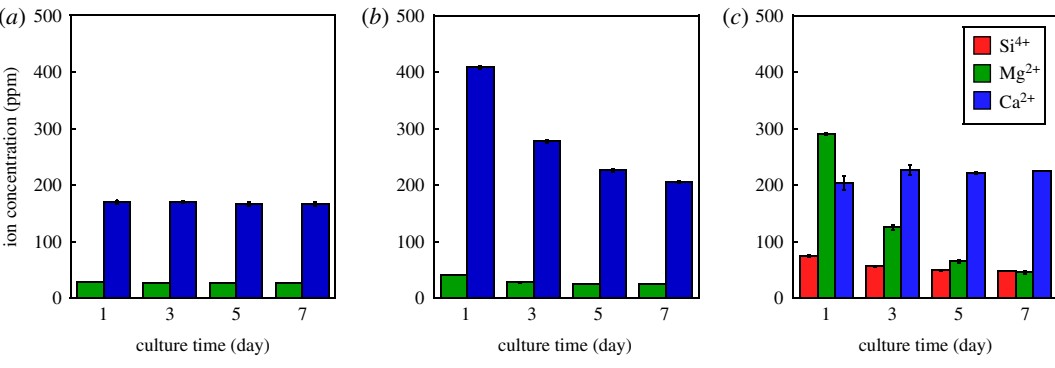

**Figure 8.** Ion-release behaviours after being cell-cultured in α-MEM from (a) TCP/PLGA, (b) V/PLGA and (c) G-V/PLGA. A culture time of '1' indicates '0–1' day, and '3,' '5', and '7' indicate '2–3', '4–5', and '6–7' days, respectively.

The Si–O–Si bond of $Q_{Si}^1$ is easily broken through hydrolysis, creating soluble silanol groups. This glass is expected to be effective as a source of release of $Mg^{2+}$ and silicate ions.

As shown in Figure 8c, when the cells were cultured in G-V/PLGA, almost all $Mg^{2+}$ ions in the G particles were released within 5 days, whereas silicate ions of 30–40 ppm d$^{-1}$ and $Ca^{2+}$ ions of approximately 50 ppm d$^{-1}$ were continuously released, except for an initial burst release within day 1. The following three factors are considered reasons for these ions-release behaviours.

The first factor is the solubility of the glass particles consisting of $Q_{Si}^1$ (73%) and $Q_{Si}^2$ (27%). The short chain structure is easily broken by $H_2O$ through a mechanism close to the reaction of a Bioglass® surface [42]. To form a silanol group on the surface, the glass releases $Mg^{2+}$ and $Na^+$ ions through an exchange with protons (or $H_3O^+$) in the medium, and a loss of soluble silica in the form of $Si(OH)_4$ will then be induced. Figure 6 indicates that, after the immersion of G-V/PLGA in the medium, the G particles are dissolved, and a thin Si-rich layer around the composite surface is formed. The layer will be a silica gel phase formed through the condensation and repolymerization of the silanols. As shown in figure 6, phosphate ions appeared within the vicinity of the sample surface after immersion in α-MEM. The silica gel layer could enhance the formation of calcium phosphate phase around the sample surface. The difference in ion-releasing behaviours between G-V/PLGA and V/PLGA in figure 8 is related to the formation of the silica layer, which inhibits the initial burst release of $Ca^{2+}$ ions from V/PLGA.

The second factor is the water-uptake capability of PLGA. PLGA has been reported to show a much higher water-uptake ability than PLLA when it contains 20% of poly(glycolic acid) and when PLLA is immersed in a phosphate buffer solution [23]. Owing to this ability of PLGA, the glass particles will be dissolved by water invading inside the material. Pathways through which the ions are released are created by this entrapped water. As a result, $Mg^{2+}$ and $Na^+$ ions are quickly released. Silicate ions are released slowly, however, owing to their low solubility, and some remain around the surface of the composite. V particles are also dissolved in the uptake water, and $Ca^{2+}$ ions are continuously released.

The third factor is the high content of the filler, namely, 60 wt%, in the composite. In our previous study, the ion-release behaviours of the PLLA composites containing 30 and 60 wt% of SiV in a Tris buffer solution were examined [32]. The composite containing 30 wt% of SiV released approximately 60% of the silicate ions within 1 day of immersion, whereas the composite containing 60 wt% of SiV released one almost completely within 1 day. This burst release is considered to be from the percolation effect of the filler, namely SiV, owing to its contact in the PLLA matrix phase. Such an effect occurs because G-V/PLGA contains 60 wt% (approx. 50 vol%) of filler (G and V particles).

The combination of these three factors indicates that water can easily infiltrate the G-V/PLGA. The high solubility of glass implies that the infiltrated water will easily dissolve the $Mg^{2+}$ and silicate ions, even if present in a small amount, namely, only 4 wt%. The large amount of V particles, namely, 50 wt%, indicates a continuous supply of $Ca^{2+}$ ions. The proliferation of MC3T3-E1 on G-V/PLGA showed significantly higher values during each culture period. As shown in figure 8, the release of $Ca^{2+}$ ions from V/PLGA, and the release of silicate, $Mg^{2+}$ and $Ca^{2+}$ ions from G-V/PLGA, was observed.

The combination of $Mg^{2+}$, $Ca^{2+}$ and silicate ions released from G-V/PLGA might enhance the proliferation of the cells. It is difficult at this stage to conclude the enhancing effect and its mechanism. However, we believe that the present study demonstrates that the inclusion of a small amount of ion-releasing glass particles might have a positive effect on the stimulation of osteoblast-like cells.

# 5. Conclusion

In this study, we aimed at the preparation of biodegradable polymer composites for tuning the release of silicate, $Mg^{2+}$ and $Ca^{2+}$ ions using highly soluble glass and calcium carbonate particles.

A $40SiO_2–40MgO–20Na_2O$ glass with high solubility was newly prepared as the releasing source of $Mg^{2+}$ and silicate ions. The glass consisted predominantly of a $Q^1_{Si}$ group with a portion of a $Q^2_{Si}$ group; this structure was shown to ease the dissolution in an aqueous solution.

A total of 4 wt% of glass particles and 56% calcium carbonate (vaterite) particles containing a trace amount of calcite were embedded in a PLGA matrix phase. The resulting composite showed a unique ion-releasing behaviour in a culture medium, demonstrating a burst release of $Mg^{2+}$ ions immediately after immersion in the medium and the continuous release of $Ca^{2+}$ and silicate ions.

Such an ion dissolution behaviour is thought to be caused by the following three factors: (i) the high solubility of the glass particles, (ii) the high water-uptake capability of PLGA and (iii) the percolation effect of the filler (60 wt%) embedded in the polymer matrix.

The continuous release of silicate ions was considered to originate from the balance between the amounts of soluble silica and the gelating one derived from the hydrolysis of the glass particles. The continuous dissolution of $Ca^{2+}$ ions originated from the high solubility of vaterite and a percolation effect with a large amount of filler, namely, 60 wt%, in addition to the hydrous capability of PLGA.

A tuning of the ion-release behaviour when using a small amount of glass particles demonstrated the possibility of enhancing the proliferation of osteoblast-like cells.

Data accessibility. The datasets supporting this article have been uploading as part of the electronic supplementary material.

Authors' contributions. N.O. and T.K. conceived of and designed the study work. A.T. prepared all samples, and acquired and analysed the data. H.M., A.O. and Y.N. analysed and interpreted data. All authors wrote the manuscript and gave final approval for its publication.

Competing interests. The authors declare we have no competing interests.

Funding. The present work was supported in part by KAKENHI 16K14403 and NEDO project supporting the advancement of strategic core technologies.

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
