## [Reviewer comments · Royal Society Open Science]

Review History

RSOS-190612.R0 (Original submission)

Review form: Reviewer 1

Is the manuscript scientifically sound in its present form?

Yes

Are the interpretations and conclusions justified by the results?

Yes

Is the language acceptable?

Yes

Is it clear how to access all supporting data?

Not Applicable

Do you have any ethical concerns with this paper?

No

Have you any concerns about statistical analyses in this paper?

No

Recommendation?

Accept with minor revision (please list in comments)

Comments to the Author(s)

The manuscript "Tuning of ion-release capability from bio-ceramic-polymer composites for enhancing cellular activity" describes the synthesis and osteoblast-material interactions. The authors build on previous findings to synthesise a novel glass/calcium carbonate/PLGA composite films and investigate ion release from these and their effect on osteoblasts. They successfully control the release rate of silicate, Mg^{2+} , and Ca^{2+} ions from these composites through clever chemistry, compositions and morphologies.

1) Check content for flow and continuity. Fix typos, e.g., page 4 line 16, □-MEM,

2) Should reference literature to confirm the deconvoluted peaks identified in ^{29}Si MAS NMR are correct for 'G'. The peaks seem to have a high chemical shift. Also, significant amount of Si are Q1 so difficult to form a glass. Was XRD performed on 'G' to check a glass was formed and if two different networks exist?

3) Is there any evidence of Si-O-Mg or Mg-O-Mg networks forming in G? This would help explain the dissolution of the glass in media.

4) Figure 6. TEM/EDX images do not indicate the elements mapped. Indicate either on the maps or in the legend what the different colours corresponds to. It seems from the main text that the green is Ca and red is Si. It would also be useful to have a schematic showing where from the composite film the lamella for STEM was taken. Brightness and contrast on Fig 6e should be adjusted. Was phosphorus mapped? It may be possible that G leads to a HCA layer formation as it dissolves locking some of the Ca^{2+} within the film.

5) The authors discuss 3 factors contributing to the ion release from the composite in media. First factor, the silica rich layer is formed on the composite films immersed in media for 7 days is suggested to inhibit Ca^{2+} release. This inhibition will be a time dependent effect. I.e., At D1 the gel layer may be thick hence higher inhibition and at D7 gel layer may be thinner hence inhibit less. TEM images and EDX maps after immersion for 1 day may help strengthen this point.

6) Vaterite transforms to calcite very rapidly in water; could the dissolving Mg^{2+} and Na^{+} from G interfere with this transformation hence slowing down Ca^{2+} release from the composite films?

7) The second factor explains the prolonged release of silicate ions while Mg^{2+} exhibits a burst release profile. Could this also be due to the preferential dissolution of a Mg-O-Mg phase in the G as well as the solubility, size and charge of silicate and Mg^{2+} ? Perhaps reporting Na^{+} concentrations in dissolution media may shed some light on this.

Review form: Reviewer 2**Is the manuscript scientifically sound in its present form?**

Yes

Are the interpretations and conclusions justified by the results?

Yes

Is the language acceptable?

Yes

Is it clear how to access all supporting data?

Yes

Do you have any ethical concerns with this paper?

No

Have you any concerns about statistical analyses in this paper?

No

Recommendation?

Accept with minor revision (please list in comments)

Comments to the Author(s)

Interesting paper; some comments for improvement are listed below:

1. This paper on ion release from bioglasses should be cited:
Bioactive glasses entering the mainstream. *Drug Discovery Today* 2018;23:1700-1704.
2. Ion release plot rather than bar chart is preferable to illustrate the trend.
3. How can a-MEM mimic body fluids? Kokubo's SBF is used for this purpose –please comment on that.

Decision letter (RSOS-190612.R0)

30-Jul-2019

Dear Mr Osada

On behalf of the Editors, I am pleased to inform you that your Manuscript RSOS-190612 entitled "Tuning of ion-release capability from bio-ceramic-polymer composites for enhancing cellular activity" has been accepted for publication in *Royal Society Open Science* subject to minor revision in accordance with the referee suggestions. Please find the referees' comments at the end of this email.

The reviewers and handling editors have recommended publication, but also suggest some minor revisions to your manuscript. Therefore, I invite you to respond to the comments and revise your manuscript.

- **Ethics statement**

- **Data accessibility**

It is a condition of publication that all supporting data are made available either as supplementary information or preferably in a suitable permanent repository. The data accessibility section should state where the article's supporting data can be accessed. This section should also include details, where possible of where to access other relevant research materials such as statistical tools, protocols, software etc can be accessed. If the data has been deposited in an external repository this section should list the database, accession number and link to the DOI

for all data from the article that has been made publicly available. Data sets that have been deposited in an external repository and have a DOI should also be appropriately cited in the manuscript and included in the reference list.

If you wish to submit your supporting data or code to Dryad (<http://datadryad.org/>), or modify your current submission to dryad, please use the following link:
<http://datadryad.org/submit?journalID=RSOS&manu=RSOS-190612>

- **Competing interests**

- **Authors' contributions**

- **Acknowledgements**

- **Funding statement**

Because the schedule for publication is very tight, it is a condition of publication that you submit the revised version of your manuscript before 08-Aug-2019. Please note that the revision deadline will expire at 00.00am on this date. If you do not think you will be able to meet this date please let me know immediately.

on behalf of Dr Maria Charalambides (Associate Editor) and R. Kerry Rowe (Subject Editor)
 openscience@royalsociety.org

Reviewer comments to Author:
 Reviewer: 1

Comments to the Author(s)

The manuscript "Tuning of ion-release capability from bio-ceramic-polymer composites for enhancing cellular activity" describes the synthesis and osteoblast-material interactions. The authors build on previous findings to synthesise a novel glass/calcium carbonate/PLGA composite films and investigate ion release from these and their effect on osteoblasts. They successfully control the release rate of silicate, Mg²⁺, and Ca²⁺ ions from these composites through clever chemistry, compositions and morphologies.

- 1) Check content for flow and continuity. Fix typos, e.g., page 4 line 16, □-MEM,
- 2) Should reference literature to confirm the deconvoluted peaks identified in ²⁹Si MAS NMR are correct for 'G'. The peaks seem to have a high chemical shift. Also, significant amount of Si are Q1 so difficult to form a glass. Was XRD performed on 'G' to check a glass was formed and if two different networks exist?
- 3) Is there any evidence of Si-O-Mg or Mg-O-Mg networks forming in G? This would help explain the dissolution of the glass in media.
- 4) Figure 6. TEM/EDX images do not indicate the elements mapped. Indicate either on the maps or in the legend what the different colours corresponds to. It seems from the main text that the green is Ca and red is Si. It would also be useful to have a schematic showing where from the composite film the lamella for STEM was taken. Brightness and contrast on Fig 6e should be adjusted. Was phosphorus mapped? It may be possible that G leads to a HCA layer formation as it dissolves locking some of the Ca²⁺ within the film.
- 5) The authors discuss 3 factors contributing to the ion release from the composite in media. First factor, the silica rich layer is formed on the composite films immersed in media for 7 days is suggested to inhibit Ca²⁺ release. This inhibition will be a time dependent effect. I.e., At D1 the gel layer may be thick hence higher inhibition and at D7 gel layer may be thinner hence inhibit less. TEM images and EDX maps after immersion for 1 day may help strengthen this point.
- 6) Vaterite transforms to calcite very rapidly in water; could the dissolving Mg²⁺ and Na⁺ from G interfere with this transformation hence slowing down Ca²⁺ release from the composite films?
- 7) The second factor explains the prolonged release of silicate ions while Mg²⁺ exhibits a burst release profile. Could this also be due to the preferential dissolution of a Mg-O-Mg phase in the G as well as the solubility, size and charge of silicate and Mg²⁺? Perhaps reporting Na⁺ concentrations in dissolution media may shed some light on this.

Reviewer: 2

Comments to the Author(s)

Interesting paper; some comments for improvement are listed below:

1. This paper on ion release from bioglasses should be cited:
Bioactive glasses entering the mainstream. *Drug Discovery Today* 2018;23:1700-1704.
2. Ion release plot rather than bar chart is preferable to illustrate the trend.
3. How can a-MEM mimic body fluids? Kokubo's SBF is used for this purpose –please comment on that.

Author's Response to Decision Letter for (RSOS-190612.R0)

See Appendix A.

Decision letter (RSOS-190612.R1)

14-Aug-2019

Dear Mr Osada,

I am pleased to inform you that your manuscript entitled "Tuning of ion-release capability from bio-ceramic-polymer composites for enhancing cellular activity" is now accepted for publication in Royal Society Open Science.

Kind regards,
Lianne Parkhouse
Editorial Coordinator
Royal Society Open Science
openscience@royalsociety.org

on behalf of Dr Maria Charalambides (Associate Editor) and R. Kerry Rowe (Subject Editor)
openscience@royalsociety.org

Follow Royal Society Publishing on Twitter: [@RSocPublishing](https://twitter.com/RSocPublishing)

Appendix A

Dear Editor,

Thank you very much for your kind letter and comments from the reviewers about our manuscript entitled “Tuning of ion-release capability from bio-ceramic-polymer composites for enhancing cellular activity”. These comments are all valuable and very helpful for revising and improving quality of the manuscript, as well as the important guiding significance to us. We have studied the reviewers’ comments carefully and revised the relevant parts in the manuscript according to these comments, and all of the questions were answered. In a revised ms, the improved portions were yellow-highlighted. Here is the list of changes:

To the comments by Reviewer #1

	Comment	Our response
1	Check content for flow and continuity. Fix typos, e.g., page 4 line 16, α -MEM,	We have checked the content and errors in detail. (i) p. 4, L. 16: Strange character was corrected. (ii) The title in Section 3.2 was revised as follows: (Old ms) 3.2. Preparation of SiV particles → (Revised ms) 3.2. Preparation of vaterite particles
2	Should reference literature to confirm the deconvoluted peaks identified in ^{29}Si MAS-NMR are correct for ‘G’. The peaks seem to have a high chemical shift. Also, significant amount of Si are Q ¹ so difficult to form a glass. Was XRD performed on ‘G’ to check a glass was formed and if two different networks exist?	(i) The reviewer has pointed out that the chemical shift may higher slightly, but the shifts are comparable to those shown in Ref. 39. So, the reference was cited in the 2 nd paragraph in section 4.1. (ii) Thank you for your great comment. We have measured the XRD pattern of G, after the reviewer’s comment, And the pattern of the ”G” was added as Figure 1(b) in the revised ms. Interestingly, the pattern showed two halo peaks. Along with this result, the following yellow-highlighted sentence was added. (Section 3.1; 2nd paragraph, in Revised ms) Figure 1(a) shows an image of the G particles using scanning electron microscopy (SEM) (JSM-6301F, JEOL, Japan). A conductive coating treatment was carried out prior to the observation using an osmium coater (NEOC Neo Osmium Coater, Meiwafoysis Co., Ltd., Japan). The glass particle sizes were approximately 1 μm or less. As shown in

		Figure 1(b), halo peaks were observed in an x-ray diffraction (XRD; X'pert, Philips, the Netherlands: CuKα, 45 kV, 40 mA) pattern of the particles; G was concluded to be glassy particles. In this pattern, two halo peaks were shown; this glass may contain two different networks. Figure caption was revised as follows: (Old ms) Figure 1 SEM image of G particles. → (Revised ms) Figure 1 (a) SEM image and (b) XRD pattern of G particles.
3	Is there any evidence of Si-O-Mg or Mg-O-Mg networks forming in G? This would help explain the dissolution of the glass in media.	In this work, it is difficult to find the direct evidence on the formation of Si-O-Mg and/or Mg-O-Mg. However, in XRD pattern (new Figure, Fig. 1 (b)), two halo peaks were observed. This may imply the existence of, at least, two types of network structures. And also, the following report describes the possibility of the MgO₄ units as network formers. Therefore, we insert the yellow-highlighted sentence into the 1st paragraph in Section 5 (Discussion). (Section 5; 1st paragraph, in Revised ms) G included approximately 70% of Q_{Si}^1 and 30% of Q_{Si}^2. In general, it may be difficult to prepare glasses containing almost no Q_{Si}^2, Q_{Si}^3, or Q_{Si}^4. Figure 1(b) may imply two types of network structures in the glass. It has been reported that, in 49.5SiO₂-1.1P₂O₅-23.0MgO-26.4Na₂O glass, ~85% of MgO act as network modifiers and ~15% as network formers, MgO₄ units [41]. Because MgO, which is an intermediate oxide, is contained in large amounts in G, some of them are considered to act as a member in glass network formers facilitating the vitrification. [Ref #41] Watts SJ, Hill RG, O'Donnell MD, Law RV. 2010. Influence of magnesia on the structure and properties of bioactive glasses. J. Non-Cryst. Solids 356, 517-524.

4	Figure 6 TEM/EDX images do not indicate the elements mapped. Indicate either on the maps or in the legend what the different colours corresponds to. It seems from the main text that the green is Ca and red is Si. It would also be useful to have a schematic showing where from the composite film the lamella for STEM was taken. Brightness and contrast on Fig 6e should be adjusted. Was phosphorus mapped? It may be possible that G leads to a HCA layer formation as it dissolves locking some of the Ca^{2+} within the film.	(i) As a result of our recheck, Figure 6 were revised. b and f, which are superimposed mapping of Ca and Si, were deleted and P map in the result after immersion was added newly, following the reviewer's comment. Since Mg was difficult to be detected clearly, it was not shown here. The elements were inserted in the maps. (ii) The sample observed in STEM was embedded in resin and then it was processed using FIB. So, to clarify the sample position, the position of "Resin" was shown in STEM photos before and after immersion. (iii) We have tried to adjust the brightness and contrast on Fig 6, but unfortunately, the contrast of the original photo was too high to be adjusted. Please accept this level of this photo. (iv) In the old ms, a P map was not shown, but, following the reviewer's comment, the map after immersion was added. The map showed the Si and P were around the sample surface on V particles. Calcium phosphate layer containing silica may form around the surface. This fact was shown in the yellow-highlighted sentences into the 3rd paragraph in Section 4.2, and the 3rd paragraph in Section 5. (Section 4.2; 3rd paragraph, the last sentence in Revised ms) ... In contrast, from the images taken after 7 days of immersion, no G particles were observed inside the material, and Si and P appeared within the vicinity of the sample surface on the V particles. (Section 5; 3rd paragraph, in Revised ms) ...The layer will be a silica gel phase formed through the condensation and repolymerization of the silanols. As shown in Fig. 6, phosphate ions appeared within the vicinity of the sample surface after immersion in α-MEM. The silica gel layer could enhance the formation of calcium phosphate phase around the sample surface. The difference in ion-releasing behaviors between G-V/PLGA and V/PLGA in Fig. 8 is related to the formation of the silica layer, which inhibits
---	--	---

		the initial burst release of Ca²⁺ ions from V/PLGA. Figure caption was revised as follows: (Old ms) Figure 6 Element mapping image of cross-sectional G-V/PLGA before and after immersion in α-MEM: (a–d) before immersion and (e–h) after 7 days of immersion. The arrow indicates the surface of the composite. → (Revised ms) Figure 6 Element mapping image of cross-sectional G-V/PLGA before and after 7 days of immersion in α-MEM. The samples were prepared by a FIB processing after being embedded in “Resin”. The arrow indicates the surface of the composite.
5	The authors discuss 3 factors contributing to the ion release from the composite in media. First factor, the silica rich layer is formed on the composite films immersed in media for 7 days is suggested to inhibit Ca²⁺ release. This inhibition will be a time dependent effect. i.e., at D1 the gel layer may be thick hence higher inhibition and at D7 gel layer may be thinner hence inhibit less. TEM images and EDX maps after immersion for 1 day may help strengthen this point.	We have no data on EDS mapping of day-1. However, the each release amount of calcium during 2 days, i.e., “day 2-3”, “day 4-5” and “day 6-7”, was almost unchanged. That is, the ion was constantly released from the composite, despite the formation of silica gel layer. The revised Fig. 6 (including P map) implies the enhancing effect of calcium phosphate phase around the silica gel layer. As the reviewer pointed out, G leads to a HCA layer formation as it dissolves, locking some of the Ca²⁺ within the film. As a result, the formation of silica gel layer would be origin of the inhibiting effect of Ca burst release. From these view points, as described in the reply to the reviewer’s comment #4, the following yellow-highlighted sentences were inserted into the 3rd paragraph in Section 5. (Section 5; 3rd paragraph, in Revised ms) ...The layer will be a silica gel phase formed through the condensation and repolymerization of the silanols. As shown in Fig. 6, phosphate ions appeared within the vicinity of the sample surface after immersion in α-MEM. The silica gel layer could enhance the formation of calcium phosphate phase around the sample surface. The difference in ion-releasing

		behaviors between G-V/PLGA and V/PLGA in Fig. 8 is related to the formation of the silica layer, which inhibits the initial burst release of Ca²⁺ ions from V/PLGA.
6	Vaterite transforms to calcite very rapidly in water; could the dissolving Mg²⁺ and Na⁺ from G interfere with this transformation hence slowing down Ca²⁺ release from the composite films?	The solubility of calcite is lower than that of vaterite. So, the release of Ca²⁺ ion should be reduced. Therefore, it is unlikely that the remaining vaterite slows down the release of Ca²⁺ ion.
7	The second factor explains the prolonged release of silicate ions while Mg²⁺ exhibits a burst release profile. Could this also be due to the preferential dissolution of a Mg-O-Mg phase in the G as well as the solubility, size and charge of silicate and Mg²⁺? Perhaps reporting Na⁺ concentrations in dissolution media may shed some light on this.	There is no evidence for the presence of Mg-O-Mg bond in this glass G. On the other hand, the existence of Si-O-Mg bond has been reported in many references, such as Ref. 41. In addition, in glass, MgO works as an intermediate oxide and is believed to form a network by assisting Si-O-; it may be difficult to form a Mg-O-Mg bond. The difference in the releasing behavior between Mg ions and silicate ions is believed to be simply due to the difference in their solubility, as described in our ms. As Na⁺ ion is contained in large amounts in α-MEM, it was almost impossible to monitor the behavior of a small amount of Na⁺ ion released from the composite containing 4wt% G. Therefore, we would like to emphasize only that Mg is one of the members in glass network formers, through inserting the yellow-highlighted phrase into the last sentence in the 1st paragraph in Section 5 (Discussion). (Section 5; 1st paragraph, in Revised ms) ...Because MgO, which is an intermediate oxide, is contained in large amounts in G, some of them are considered to act as a member in glass network formers facilitating the vitrification.

To the comments by Reviewer #2

	The comment	Our response
1	This paper on ion release from bioglasses should be cited: Bioactive glasses entering the mainstream. Drug Discovery Today 2018;23:1700-1704.	Following the reviewer's comment, the article was cited in page 2, Line 28, as Ref. 36. [36] Kargozar S, Baino F, Hamzehlou S, Hill RG, Mozafari M. 2018. Bioactive glasses entering the mainstream. Drug Discovery Today. 23, 1700-1704. (doi:10.1016/j.drudis.2018.05.027)
2	Ion release plot rather than bar chart is preferable to illustrate the trend.	In the ion dissolution test in α-MEM, since α-MEM was exchanged at day 1, 3 and 5. This is because the ion amounts were measured according to the condition of cell culture test. That is, the results do not show the continuous ion release. Therefore, we believe that these graphs are better to be represented using the bar graph. Our intention is indicated in Figure captions being showed as "An immersion time of 1" means "0-1" day, where as "3," "5," and "7" indicate "2-3," "4-5," and "6-7" days, reflectively". In order to make it clearer further, we added the following yellow-highlighted sentence into the 1st paragraph in Section 3.4. (Section 3.4; 1st paragraph, in Revised ms) ...with 5% CO₂ for 7 days. The medium was changed after 1 day of culturing and then changed every other day.
3	How can α-MEM mimic body fluids? Kokubo's SBF is used for this purpose –please comment on that.	In this work, in order to consider the result of the ions released from the composite as well as that of the cell culture test, the dissolution behavior to α-MEM, which is used for culture test, was used.

(Over)

We hope this revised ms is satisfactory for accepting for publication.

Sincerely yours,

Naoki Osada and Toshihiro Kasuga

(Corresponding authors)